# Ensuring robustness in scientific research, split-root assays as an example case

Lucila Salvatore[1,2], Ronald Pierik[1,3], Kaisa Kajala[1] and Kirsten ten Tusscher[1,2]

[1]Experimental and Computational Plant Development group, Utrecht University, Utrecht, The Netherlands; [2]Theoretical Biology group, Utrecht University, Utrecht, The Netherlands; [3]Laboratory of Molecular Biology, Wageningen University and Research, Wageningen, The Netherlands

**Perspective**

nitrogen; replicability; reproducibility; robustness; split-root.

**Corresponding authors:**
Lucila Salvatore and Kirsten ten Tusscher;
Emails: l.salvatore@uu.nl,
K.H.W.J.tentusscher@uu.nl

**Associate Editor:**
Dr. Christian Fleck

## Abstract

Scientific progress relies on reproducibility, replicability, and robustness of research outcomes. After briefly discussing these terms and their significance for reliable scientific discovery, we argue for the importance of investigating robustness of outcomes to experimental protocol variations. We highlight challenges in achieving robust, replicable results in multi-step plant science experiments, using split-root assays in Arabidopsis thaliana as a case study. These experiments are important for unraveling the contributions of local, systemic and long-distance signalling in plant responses and play a central role in nutrient foraging research. The complexity of these experiments allows for extensive variation in protocols. We investigate what variations do or do not result in similar outcomes and provide concrete recommendations for enhancing the replicability and robustness of these and other complex experiments by extending the level of detail in research protocols.

## 1. Introduction

While the general public may imagine scientific progress to arise from isolated sparks of genius of individual scientists, science to a large extent is an incremental and collaborative endeavor. For science to progress in an efficient manner time- and resource-wise, it is critical that results can be repeated and built upon. The apparent lack of reproducibility and repeatability of results both inside and outside of the life sciences has received considerable interest over the last decade (Baker, 2016; Hicks, 2023) and has resulted in a strong focus on making articles, protocols, analyses, simulation codes, and results publicly available and reproducible. While FAIR principles are nowadays broadly followed (Jacobsen et al., 2020; Wilkinson et al., 2016), recreating results from complex protocols is far from trivial, and determining the causal factor underlying different outcomes can be a substantial burden.

In scientific research, the term *reproducibility* is typically reserved for the capacity to generate quantitatively identical results when using the same methods and conditions. For data analysis and computational biological research, in theory, full reproducibility can be achieved if data, analysis protocols, and codes are available. In practice, limited documentation on the settings of codes and preprocessing of data may still hamper this reproducibility. In experimental research, generating identical results on biological systems even in the same lab with the same equipment, conditions, and person executing the experiment is highly unlikely due to noise from both biological sources and experimental execution. Therefore, under the Claerbout/Donoho/Peng convention, in experimental biology we speak of *replicability* when experiments, performed under the same conditions, produce quantitatively and statistically similar results (Barba, 2018). Still, even striving for replicability rather than reproducibility can be a daunting task, sometimes even resulting in the somewhat ominously sounding phrase 'in our hands' in some papers.

One may think that striving for replicability and reproducibility is sufficient to collect the best quality data and gain new knowledge in the most efficient possible manner. However, *robustness* of outcomes may also inform us about the significance of the biological phenomenon we are observing. What we define as robustness here, for experimental biology, is the capacity to generate similar outcomes also in slightly different conditions (Kitano, 2004). In computational biology, where we create models to simulate biological phenomena of interest, it is common

**Table 1.** Comparison of split-root assays used to investigate nitrogen foraging in *Arabidopsis thaliana*

| Paper | HN concentration | LN concentration | Photoperiod-light intensity (mmol m$^{-2}$ s$^{-1}$) | Days before cutting | Recovery period | Days on heterogenous treatment | N source in the growth media | Sucrose concentration | Temp |
|---|---|---|---|---|---|---|---|---|---|
| Ruffel et al. (2011) | 5 mM KNO$_3$ | 5 mM KCl | Long day– 50 | 8–10 days | 8 days | 5 days | 0.5 mM NH$_4^+$-succinate and 0.1 mM KNO$_3$ | 0.3 mM | 22°C |
| Remans et al. (2006) | 10 mM KNO$_3$ | 0.05 mM KNO$_3$ + 9.95 mM K$_2$SO$_4$ | Long day– 230 | 9 days | None | 5 days | 10 mM KNO$_3$ | None | 22°C |
| Poitout et al. (2018) | 1 mM KNO$_3$ | 1 mM KCL | Short day– 260 | 10 days | 8 days | 5 days | 0.5 mM NH$_4$-succinate and 0.1 mM KNO$_3$ | 0.3 mM | 22°C |
| Girin et al. (2010) | 10 mM NH$_4$NO$_3$ | 0.3 mM KNO | Long day– 125 | 13 days | None | 7 days | Not specified | 1% | 21°C / 18°C |
| Tabata et al. (2014) | 10 mM KNO$_3$ | 10 mM KCl | Long day– 40 | 7 days | 4 days | 5 days | 10 mM KNO$_3$ | 0.5% | 22°C |
| Mounier et al. (2014) | 10 mM KNO$_3$ | 0.05 mM KNO$_3$ + 9.95 mM K$_2$SO$_4$ | Long day– 230 | 6 days | 3 days | 6 days | Not specified | Not specified | 22°C |
| Ohkubo et al. (2017) | 1 mM KNO$_3$ | 10 mM KCl | Photoperiod not specified– 50 | 7 days | 4 days | 5 days | N-rich medium contains 10 mM NH$_4$Cl and 10 mM KNO$_3$. | 0.5% | 22°C |

*Note:* The table shows the wide variation in nitrogen concentrations in high nitrogen (HN) and low nitrogen (LN) treatments, other media components and their concentrations, length of the protocol, duration of the treatment, light intensity and photoperiod, and temperature conditions.

practice to investigate the robustness of models to changes in parameters or model assumptions. The general idea is that for a reliable model, outcomes should only depend significantly on certain parameters, e.g. whether we simulate growth under normal versus drought conditions or the stability of a key protein yet should remain relatively constant to moderate changes in most parameters.

A robust model is more likely to simulate the right behaviour for the right reasons than a model that is sensitive to all its finely tweaked parameter values. In a similar vein, it would be informative for experimental protocols to have knowledge about which changes in the protocol have substantial effects on outcomes and which changes are buffered against. In plant biology, robust outcomes of experiments under slight variations in protocol are more likely to be outcomes that are relevant in natural conditions. This increased relevance stems from the fact that natural conditions constitute a more variable environment compared to the environment in the experiment. Additionally, experimental protocols with robust outcomes would enhance the potential for similar research to be performed in e.g. labs with less funding or different equipment, by allowing some flexibility in concentrations, time windows, or equipment used (Nosek et al., 2022). However, even if methods in papers or (e-) lab journals are fully complete, most times it is not clear whether a particular aspect of a protocol was optimized for, or if it rather results from habit or random choice. While it is not trivial to vary and investigate for each and every aspect of a protocol whether it is essential, for some aspects this information is available. Despite being available, this type of information is typically not provided in either protocols or methods sections. It is of great importance to be aware that the way we write our materials and methods and what we omit can be decisive for the success of future research projects.

Here, we will use split-root assays to investigate nutrient foraging in *Arabidopsis thaliana* as an example case. Split-root experiments are used in a wide variety of plant species and for various types of research fields and have the power to discern local from systemic responses. The main goal of these types of assays is to divide the root system architecture (RSA) into halves and expose each half to different environments. In the case of plant nutrient foraging, these studies are of major importance for unraveling the systemic signaling pathways that indicate the demand for nutrients against local supply, enabling plants to preferentially invest in root growth in locations of high nutrient supply (Hackett, 1972; Ruffel et al., 2011; Torres-Martínez et al., 2019; Zhang & Forde, 1998; Zhang et al., 1999). We will illustrate the practical problems we encountered with obtaining robust replicable results based on published methods for Arabidopsis split-root heterogeneous nitrate supply experiments. We will also suggest solutions and recommendations for alleviating these problems that are of wider applicability to complex experimental plant science protocols.

Split-root experiments have been applied to a wide variety of species and for a range of research questions (Saiz-Fernández et al., 2021). As a consequence, a wide variety of protocols exists, ranging from simply dividing a well-developed root system over two pots (Schortemeyer et al., 1993), to splitting the main root of plants with thick enough roots in two halves (Remans et al., 2006), and from grafting an additional main root (Kassaw & Frugoli, 2012) to cutting off the main root after two lateral roots (LRs) have developed to use these two laterals in the two different nutrient compartments (Remans et al., 2006). These protocols have been reviewed in detail elsewhere (Saiz-Fernández et al., 2021).

Even if we constrain our analysis to split-root experiments in Arabidopsis grown on agar plates used to analyze nitrate foraging, where the main root is cut away after two laterals have formed, still a large variety of experimental protocols exist (Table 1). Protocols

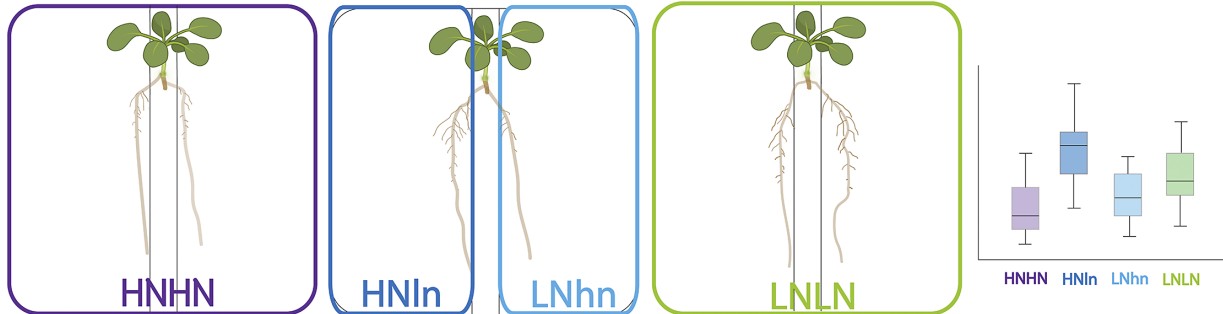

**Figure 1.** Schematic illustration of the Ruffel et al. (2011) results for split-root experiments with different nitrogen concentrations. The cartoon plates represent real plates after a nitrogen split-root assay, and the box plot represents expected results for lateral root (LR) traits such as summed LR length, number and density. The colors surrounding each plant correspond with the colors of the boxplot.

differ in the duration and number of growth steps, concentrations of high and low nitrate used, light levels, sucrose concentration in the media, and other parameters. Additionally, protocols may differ in the analysis of root growth and architecture, with some studies focusing e.g. on differences in overall root system growth between two root halves whereas others appear to focus only on specific parts of the root system.

Despite this variation, all the papers included in Table 1 robustly observe preferential foraging, i.e. the preferential investment in root growth at the side of the split-root system where the plant experiences the highest nitrate levels (HNln) (Figure 1 *HNln > LNhn*). However, in the seminal paper by Ruffel et al. (2011) it was additionally reported that in plants grown in heterogeneous nitrate, the high nitrate (HNln) side invests more in root growth compared to plants where both sides experience high nitrate (HNHN) (Figure 1 *HNln > HNHN*). Additionally, the low nitrate (LNhn) side invests less in root growth compared to roots grown in a homogeneous low nitrate (LNLN) split-root set-up (Figure 1 *LNhn < LNLN*). These results have been taken as an important hallmark for demand and supply signalling, and hence, robustness of these phenotypes may be critical to fully sample the repertoire of local, systemic, and long-distance signals involved in root foraging decisions.

Our aim was to arrive at a robust reproducible protocol reca-pitulating the Ruffel et al. (2011) results which were as simple and efficient as possible. To this end, we started with the protocol described in Figure 2 by the black typed text. As we tested the effect of various parameters, the protocol was updated as explained in the next sections.

To investigate the robustness of plant root nitrogen foraging responses to variations in protocol we will focus on three major aspects. The first is the preferential foraging response, where the root system of the high nitrogen side (HNln) should be more devel-oped than that low nitrogen side (LNhn). The second aspect is the comparison between the two homogeneous treatments, where we expect a stronger investment in root growth in LNLN plants than in HNHN plants given previous observations of foraging versus systemic repression responses (Giehl & von Wirén, 2014; Gruber et al., 2013; Ruffel et al., 2011). Finally, the roots from the HNln treatment are expected to outgrow HNHN roots, consistent with demand signalling, while the roots from the LNhn treatment are expected to be less developed compared to LNLN roots, consistent with supply signalling (Figure 1).

## 2. Results

### 2.1. A secret numbers game

Methods sections typically describe what is done to plant materials, at which time point, with which concentrations, and so on (Varapparambath et al., 2022). More extensive protocols, published as stand-alone papers additionally provide a shopping list of required materials and equipment as well as describe more extensively the sequence of steps involved in the experiment, the time they take, and potential problems and solutions (Bjornson et al., 2015; Mathew et al., 2023). Perhaps surprisingly, information on the number of seeds or plants that the experiments started with, and the percentage kept for subsequent experimental steps are only infrequently and incompletely provided (Greenwood et al., 2022; Xuan et al., 2018). These differences between the initial number of seeds and plants kept throughout the experiment may result from failure to germinate or recover, or substantial differences in development or growth. Particularly in the case of complex multi-step protocols information on these numbers is important.

To illustrate this point, based on the experience in our lab, where previously root growth and architecture phenotyping was done only on relatively young, intact root systems, typically the number of seeds used for a protocol is five times the number of plants eventually used for phenotyping. This 5-fold increase comes from the observation that on average 20% of seeds display efficient and homogeneous germination and seedling establishment, with remaining seeds failing to germinate or slowly germinating. These latter seedlings are discarded to avoid heterogeneity affecting the outcomes of our experiments. Thus, given that we required 30 plants for split-root phenotyping experiments, 8 for HN-HN, 8 for LN-LN and 14 for HN-LN combinations, we initially started with 150 seeds.

However, 150 seeds did not nearly result in 30 plants suitable for split-root experiments. Instead, due to the many steps in a split-root protocol, substantial numbers of additional plants needed to be discarded at various stages. First, for cutting the main root, only plants that have grown two LRs at the time point of cutting can be used. Alternatively, if cutting is not done below the first two LRs, but rather position-based, only plants developing two LRs after cutting can be maintained for further experiments as each half of the split-root plate requires a main root (Figure 1). In both cases, after the recovery phase, plants developing adventitious roots (AR) at the root-shoot junction and at the cutting point need to

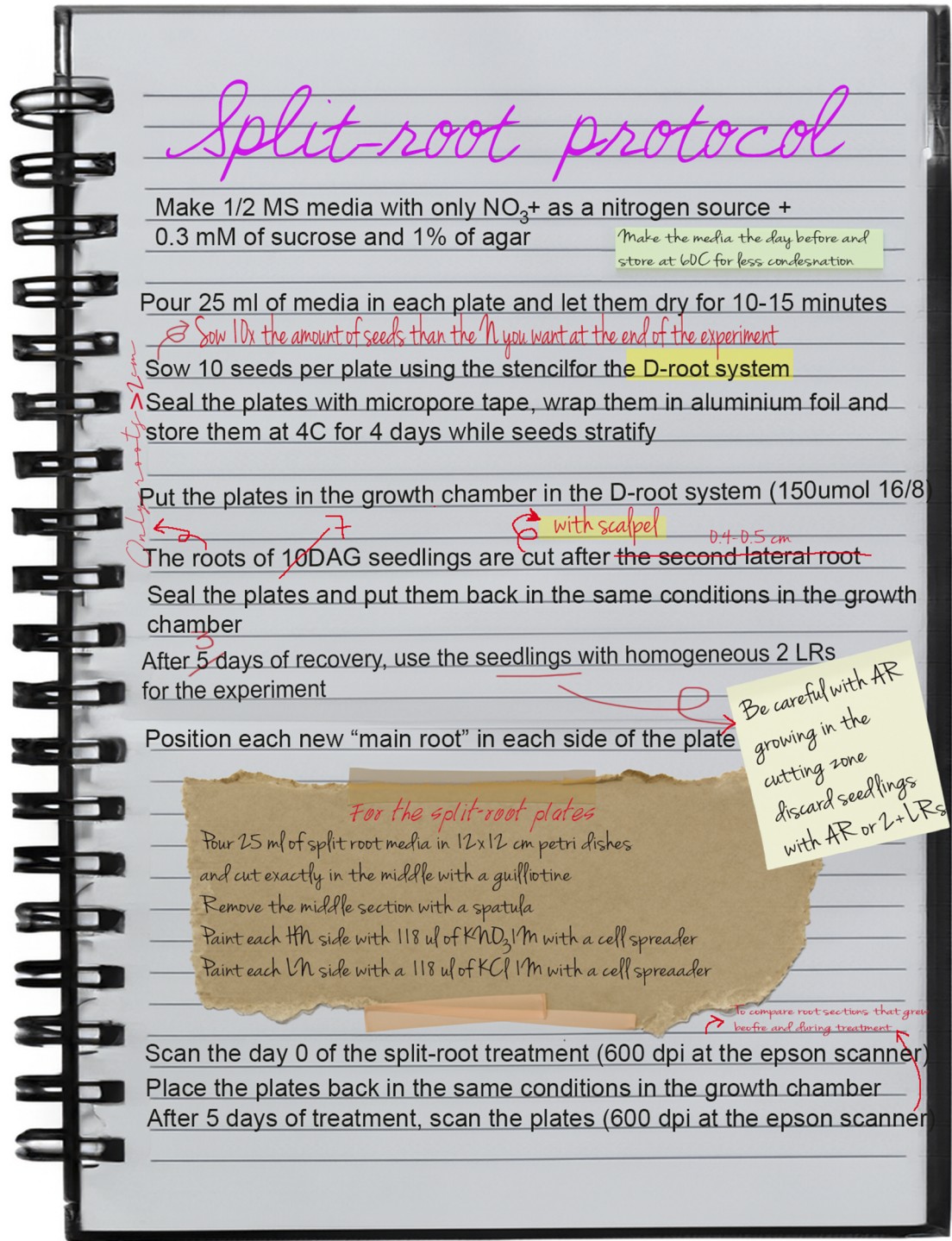

**Figure 2.** Representative experimental protocol. Typically, some steps are adjusted through repetition to achieve the most replicable outcome in the most efficient way possible. The black typed text represents the original protocol, and the adjustments are visualized as 'post-it notes' or red 'handwriting' comments and the most important of these additions are discussed in detail in the main text.

be discarded as well, as these are different types of roots and alter the root system structure (explained in section 2.4.2). Additionally, only plants with approximately similarly sized LRs can be deployed for the actual split-root treatment as otherwise, the initial size and resource differences may obscure differences in response to nutrient conditions.

After substantial trial and error and consultation with one of the authors of previously published split-root experiments, we arrived

at a protocol starting with 300 seeds. We selected this number considering two major losses. First, loss rates of 50% of the plants in the cutting step, due to insufficient development towards a large enough root, where we set the threshold to having a main root longer than 1.5 cm. Second, a loss of around 70% of the cut and recovered plants during the selection for the split-root plates, discarding here those plants that have grown either adventitious roots, extra roots, or whose two LRs are not similarly sized and/or

dissimilar from the mean LRs length in the experiment. Overall, it becomes evident that conducting a rigorously controlled experiment with homogenous plants results in a loss of approximately 90% of the seedlings started with, requiring us to start with at least 10 times the number of plants needed for final analysis. The reporting of rough estimates of the loss rates at crucial handling steps would speed up protocol adoption in a different lab.

### 2.2. Robustness and sensitivity to variations in the methodology

Not all parameters in our protocol are expected to have a significant effect on the phenotype that we are observing. On the other hand, there will also be factors in our protocol that have a major effect on the plant's response to the treatment. While it may be impossible to determine the impact of all aspects of our protocol, it may still be helpful to report for those aspects that were varied whether they influenced the observed phenotype or whether experimental outcomes were found to be robust against these variations.

To illustrate this, we will describe our experience with the split-root protocol, showing some examples of conditions that were found to have a significant effect on outcomes as well as some of them that appeared to have limited impact on the phenotypes of interest.

### 2.3. Robustness to variation in protocol parameters

**2.3.1. Seedling age during treatment.** As shown in Table 1, split-root protocols differ in the number of days at which the main root is cut, the recovery time after the cut before being transferred to a split-root medium, and even the number of transfer steps. Combined this has a substantial effect on the age and thereby the size of the seedlings at the time they are exposed to heterogeneous nitrate conditions. A lower seedling age is advantageous both in terms of protocol duration and in efficiency of root system phenotyping, as for older plants root systems become increasingly large and complex and therefore harder to phenotype. However, plant age may also impact the responses that are being studied.

We compared experimental outcomes for two different seedling ages. In the first case, we let seedlings grow for 10 days after germination (DAG), after which they were cut below the 2nd visible LR and left to recover for 5 days before exposing them to the split-root treatment. In the second case, we cut the main root of 7 DAG seedlings 0.4 cm below the hypocotyl-radicle junction point and let them recover for 3 days before exposing them to split-root treatment. Note that the rationale for now cutting below a certain distance rather than below the first two visible LRs is that, at this earlier age, two laterals are not generally visible yet. The distance of 0.4 cm was derived from observations at later time points in which results for different cutting distances were compared, indicating that in most plants two LRs would form within this 0.4 cm distance. Both younger and older seedlings were exposed to 5 days of split-root treatment before measuring the RSA.

These differences in the duration of protocol steps enabled us to compare the responses of 20 DAG or 15 DAG plants (Figure 3a,b). The average length of the LRs and the summed total length of LRs were all significantly larger in older plants, yet the responses to the treatments were highly similar in both experiments (Figure 3). Both 15 DAG (Figure 3c, Supplementary Table 1) and 20 DAG (Figure 3e, Supplementary Table 1) plants showed the same trend in total LR length as we discussed before for the overall foraging versus systemic repression (LNLN>HNHN), and supply and demand responses (HNln>HNHN, LNhn<LNLN). The same

pattern was also obtained for both plant ages when instead of the total LR length, the average LR length was analyzed (Figure 3d,e, Supplementary Table 1). This implies that the bigger total LR length observed under e.g. LNLN versus HNHN or HNln versus LNhn, was not mainly due to increased LR number but at least partly arises from individual longer roots. These results demonstrate that seedling age, at least within the range tested here, does not substantially affect the nitrogen-foraging phenotypes of interest. While trends are the same, not all comparisons show similar significance and there were fewer significant differences for the 20 DAG. There may be a difference in power of the assay depending on the length of the protocol, but establishing such a difference with certainty would require additional repetitions beyond the scope of this manuscript.

Notably, when instead number of LRs was compared between the two protocols there was only a preferential foraging phenotype for the short but not the longer protocol (Supplementary Figure S1). Considering that the whole root system was analyzed, the lack of difference in LR number may be caused by the fact that for the long protocol, the old part (where the primordia were mostly preset before the treatment) is outweighing the effect of the new part (where the primordia are being established in the context of the split-root treatment). For the short protocol, due to the shorter main root at the start of the nutrient experiment, the contribution from the new part of the root is larger.

**2.3.2. Measurement section of the root system.** Different studies use different sections of the root for analysis (Mounier et al., 2014; Remans et al., 2006; Ruffel et al., 2011). For some root traits, this may have biological relevance. For example, counting LR number/primordia in a section of the root that existed before the start of treatment is not very informative in studying a treatment response, as most root primordia in that section likely started to form prior to the treatment. In such a case, focusing on a newly formed primary root section might be more appropriate. However, this may not necessarily be the case for all root traits investigated. For example, LR length could potentially respond to nitrate, irrespective of whether these LRs arose from pre-existing or newly formed roots/primordia.

In light of these considerations about pre-existing versus newly formed tissues, we reanalyzed the nitrogen split-root 7+3 day experiment while considering three different root system compartments: the overall main root that is in the treatment (total root), the new part of the main root that grew during the treatment (new root), and the old part of the main root that grew before the transfer to the treatment (old root) (Supplementary Figure S2).

Focusing first on total LR length, in both the root segment and the whole root analyzed, we observed a preferential LR growth in HNln relative to the LNhn part of the same plate (Figure 4a–c, Supplementary Table 2). Additionally, LNhn roots had less LR growth than LNLN roots in all segments, and HNln roots had a longer total LR system than HNHN roots in the new root. Finally, LNLN had a larger root system than HNHN roots in all root segments, showing the well-known foraging response of the plant in poor nitrogen conditions. Thus, for total LR length, this set of phenotypic responses is robust to variations in the part of the root system being used for the analysis. Similarly, LR density (Supplementary Figure S3) shows a robust phenotype in the three sections as well, enabling the observation of foraging responses and a preferential phenotype in all 3 sections.

Next, we focused on average LR length. The results demonstrated a consistent phenotype for the old and total root section for both the foraging phenotype (LNLN>HNHN) and the preferential

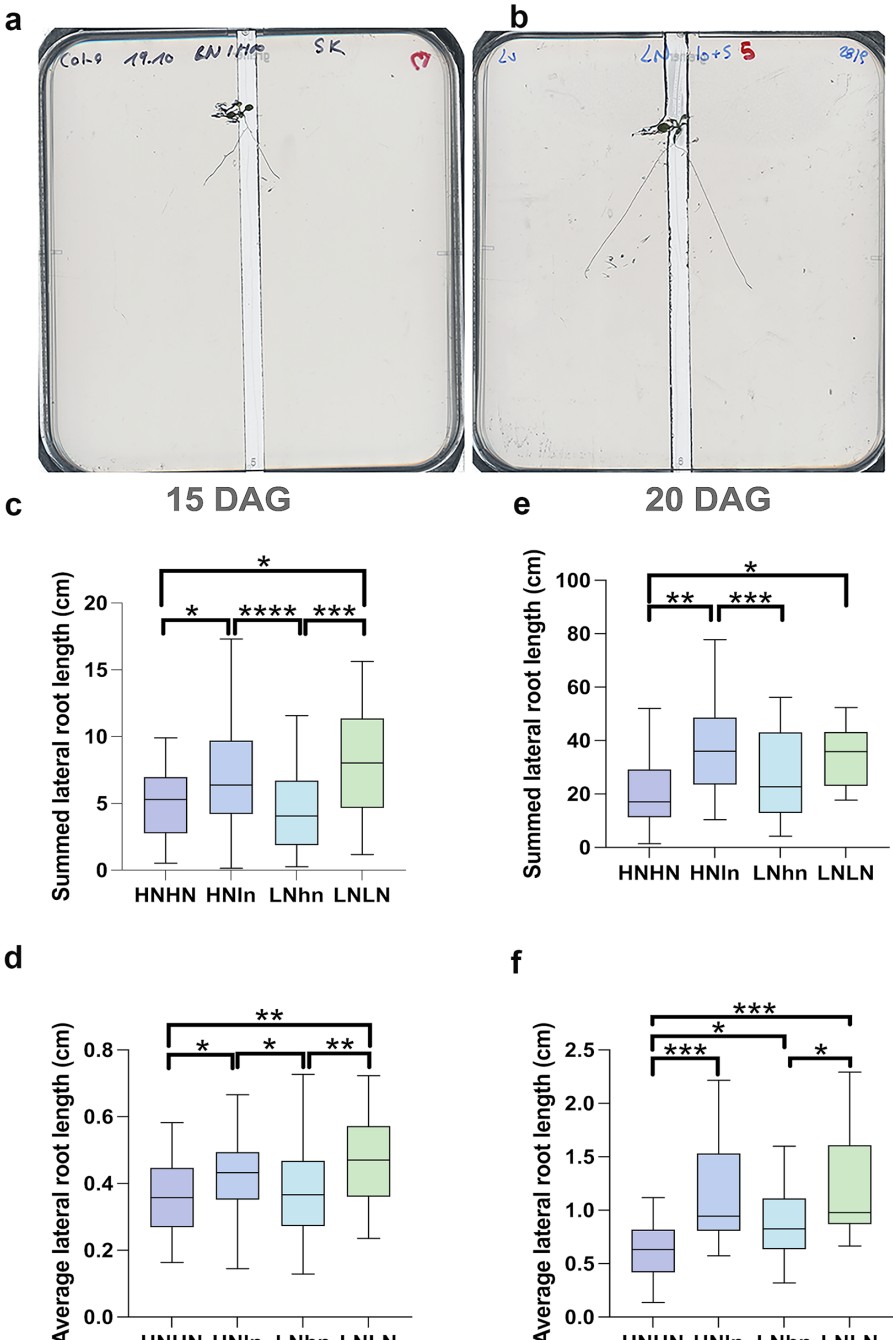

**Figure 3.** Comparison of Arabidopsis split-root phenotypes for different nitrogen availabilities in a long (20 DAG) and a short (15 DAG) protocol. *Arabidopsis thaliana* root system response after 5 days of treatment in split-root conditions in either 15-day-old seedlings (a-c-d) or in 20-day-old seedlings (b-e-f). Seedlings were grown in d-root systems the entire treatment and the cut was performed with a razor blade. Typical 10 DAG seedling at treatment day 0 in the 7+3 setup (a). Typical 15 DAG seedling at treatment day 0 in the 10+5 setup (b). LRs responses to different nitrate provision conditions: HNHN, HNLN, and LNLN, where we used for HN the 10 mM $KNO_3$ and for LN0.2 mM $KNO_3$ The summed length of the LRs of each half of each individual plant was measured (c, e), and the average length of the LRs of each half of each plant (d-f). The traits were measured for the total section of the system. Boxplot displays the median of each group ($n$ = 26-55 roots) bounded by the first and third quartiles. Asterisks indicate the significance in the comparison of two nitrogen treatments: $^*P < 0.05$; $^{**}P < 0.01$; $^{***}P<0.001$. The HNln vs LNhn split-root treatment was compared using a Wilcoxon test and the rest of the comparisons were performed using a Mann–Whitney test.

phenotype (HNln> LNhn) (Figure 4 e,f, Supplementary Table 2). However, there was no clear foraging response for the average LR length in the new section (Figure 4d). This lack of response may be attributed to the limited time available for new LRs in LNLN to achieve length differences. We hypothesize that with prolonged growth, the LRs in the new section will eventually also display significant size differences.

As expected, LR numbers (Figure 4g–i, Supplementary Table 2) showed a significant preferential root foraging phenotype in all three sections of the roots (HNln >LNhn), yet the foraging phenotype (LNLN>HNHN) could only be observed for the new part of the root. We hypothesize that primordia in the old part at least partly formed prior to the treatment start, causing reduced responsiveness in terms of LR number. As for the lack of difference

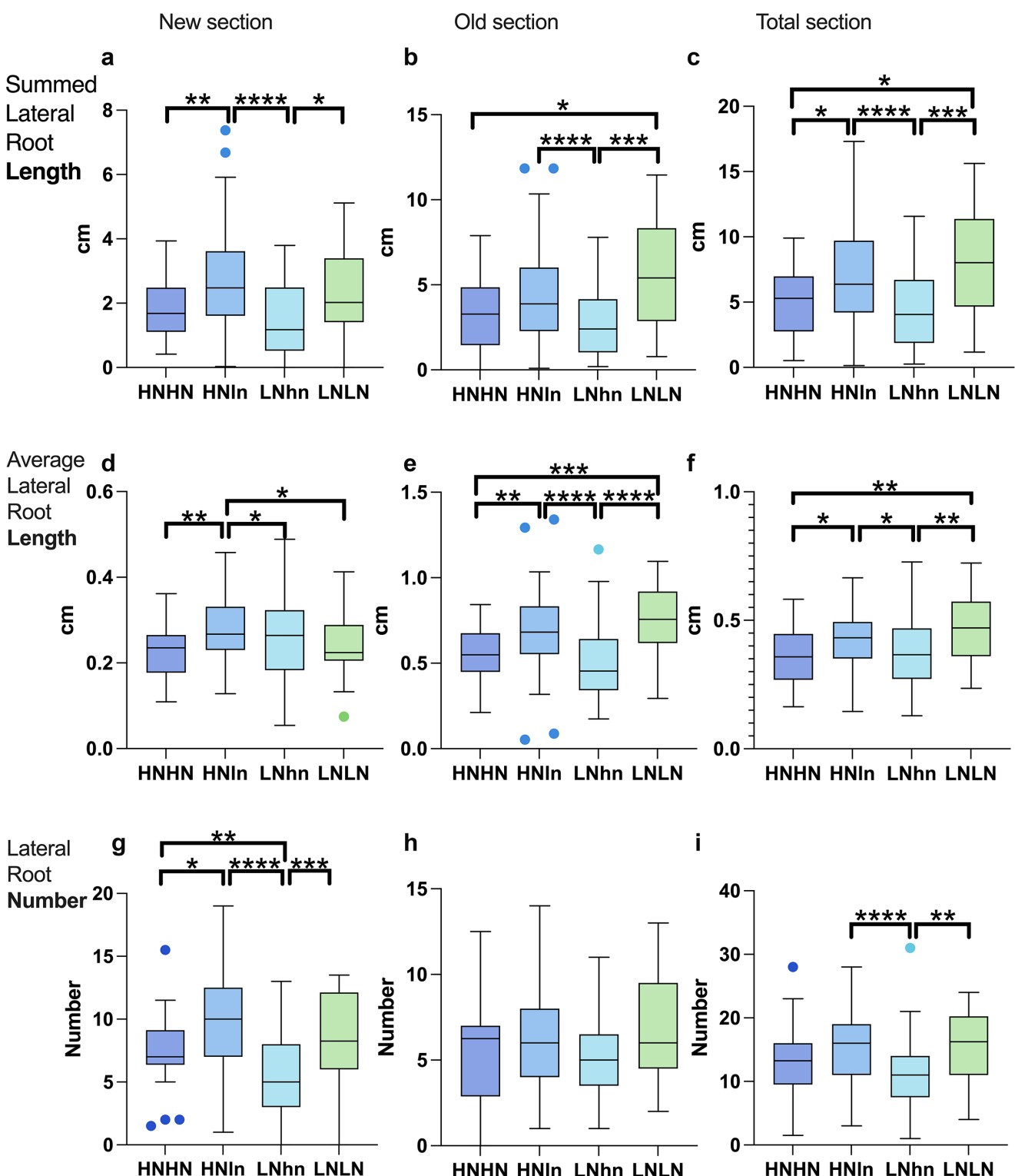

**Figure 4.** Comparison of Arabidopsis split-root root phenotypes for different nitrogen availability in different sections of the root. *Arabidopsis thaliana* root system response in 15 DAG old seedlings after 5 days of treatment in split-root conditions. Seedlings were grown in d-root systems the entire treatment and the cut was performed with a razor blade. RSA traits were measured in different root sections: New section, the part of the main root that grows in the treatment (a, d, g); Old section, the part of the main root that was already developed before the exposure to treatment (b, e, h). Total section, the whole root (c, f, i). RSA traits measured are the summed length of LRs (a–c), the average LR length (d–f), and the total number of LRs (g–i). The Boxplot displays the median of each group (*n* = 26–55 roots) bounded by the first and third quartile with individual outliers identified by the Tukey test marked as points. Asterisks indicate the significance in the comparison of two nitrogen treatments: *P < 0.05; **P < 0.01; ***P < 0.001. The HNln vs LNhn split-root treatment was compared using a Wilcoxon test and the rest of the comparisons were performed using a Mann–Whitney test.

## Light

**a**

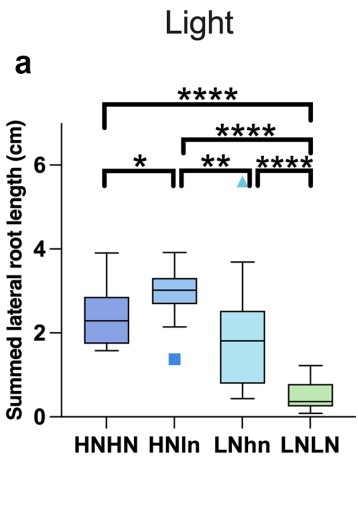

## Dark

**b**

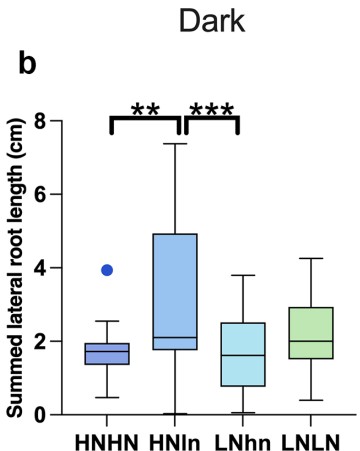

**c**

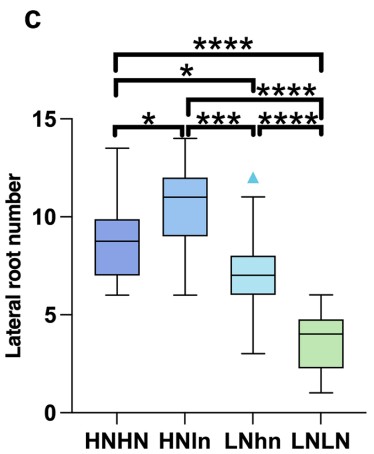

**d**

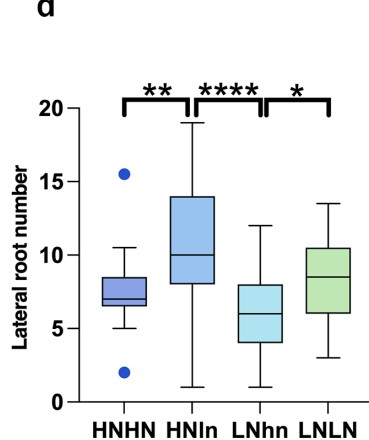

**Figure 5.** Comparison of Arabidopsis split-root phenotypes for different nitrogen availability with the roots covered or exposed to light. Arabidopsis thaliana 15 DAG old root system response after 5 days of treatment in split-root conditions in either seedlings with roots exposed to light (non-covered) (a, c) or seedlings with roots under D-root system (b, d). Seedlings were cut with a razor blade when the protocol was performed. Different RSA traits measured were the summed length of all LRs (a, b), and the number of the LRs in each plant (c, d). The traits were measured for the new section of the root system. Boxplot displays the median of each group ($n$ = 15-30 roots) bounded by the first and third quartile with individual outliers identified by the Tukey test marked as points. Asterisks indicate the significance in the comparison of the two nitrogen treatments: $^*P < 0.05$; $^{**}P < 0.01$; $^{***}P < 0.001$. The HNln vs LNhn split-root treatment was compared using a Wilcoxon test and the rest of the comparisons were performed using a Mann–Whitney test.

in LR number in the total section, we conclude that the lack of differences in the older part outweighs the differences observed in the new part of the root.

Summarizing, when focusing on LR length or LR density, results are robust to variations in the root system segment being used for analysis, while for average LR length, we found clearest responses in old root sections and total root, and for LR numbers instead we observed clearest responses in new root sections. Moreover, if only the split-root heterogeneous response is analyzed the preferential growth in the HN side compared to the LN side is prevalent in all the sections and root traits.

### 2.4. Sensitivity to variation in protocol parameters

**2.4.1. Light exposure of roots.** It has been demonstrated that some previously reported RSA responses may not represent a plant's 'natural' response but may rather be an artifact of the experimental setup, e.g. the light exposure of roots in many plate assays (Silva-Navas et al., 2015). Using the so-called D-root (dark-grown root) system, Yokawa et al. (2014) reported an enhanced halotropic response to moderate-strength salt gradients, indicating repression

of this response in light-exposed roots. In contrast, Silva-Navas et al. (2019) reported that RSA responses to phosphate starvation, increase of LR density, and severe shortening of MR are significantly stronger in light versus dark-grown roots. Thus, it is important to describe root light conditions when writing methods or protocols.

To the best of our knowledge, agar plate split-root experiments for Arabidopsis have thus far all been performed on light-exposed roots, with the experiments we described in this article thus far being the exception. To investigate whether preferential investment in HNln versus LNhn or the foraging response in LNLN versus HNHN depend on root light exposure, we compared split-root nitrogen phenotypes between experiments with the roots being covered with a D-root system and experiments where the roots were fully exposed to white light conditions (150 μmol/m$^2$s light intensity in a 16/8 light/dark growth chamber). We studied these responses in the new parts of the root system.

We observed that the preferential foraging response (HNln > LNhn) is independent of root light exposure, and can be observed for LR length, number, and density (Figure 5, Supplementary Figure S4, and Supplementary Table 3). These findings agree with

previous results showing qualitatively similar preferential foraging responses to heterogeneous nitrate in light-grown Arabidopsis plants on agar and soil-grown maize plants (Remans et al., 2006; Yu et al., 2014).

Comparing root growth under HNHN vs LNLN in dark-grown conditions to light-grown conditions, significant differences can be observed. In dark-grown roots, our results suggest a foraging response (LNLN > LNhn and HNHN) for total lateral root length, number, and density (Figure 5b,d, Supplementary Figure S4, and Supplementary Table 3). This response is expected because, in the latter two conditions, systemic nitrate levels are higher, reducing investments in root growth. In contrast, for light-exposed roots, this foraging response was no longer observed. Instead, LNLN LR length, number, and density were reduced rather than increased compared to HNln, LNhn, and HNHN conditions (Figure 5a,c, Supplementary Figure S4, and Supplementary Table 3). Thus, our experimental outcomes are partly sensitive to whether the root system is exposed or covered from light.

Furthermore, when comparing shoots of light-exposed plants for LNLN versus HNLN or HNHN plants, we saw a highly reduced rosette area in the LNLN plants (Supplementary Figure S5). While we have not investigated this in further detail, these observations suggest that the combined stress of low nitrate provision and light-exposed roots resulted in starvation rather than a foraging response, i.e. reduced instead of increased growth. The fact that this did not occur in previous studies with light-grown roots may arise from these experiments having been optimized for light conditions, e.g. through the addition of sucrose, differences in prior growth conditions, or intermediate steps, whereas we optimized our protocol for dark-grown roots.

**2.4.2. Cutting instrument.** While describing the machines used for PCR or microscopy, simpler tools used in our protocols are often given limited consideration when writing methods sections. One commonly overlooked tool is the instrument employed for cutting the plants in protocols involving e.g. split-root, grafting, or root tip regeneration. Cutting may be performed either with razor blades or a scalpel. Cutting away plant parts inflicts an injury, resulting in the activation of various stress signalling pathways (Acosta et al., 2013; Durgaprasad et al., 2019). However, different forms of damage elicit different responses. For instance, Durgaprasad et al. (2019) discussed that in the context of root tip regeneration cutting approaches resulting in sharp cuts should be preferred, as blunter cuts resulted in more damage and failure of regeneration.

In our split-root experiments, we frequently observed the formation of AR at the site where the main root was cut. In *A. thaliana*, ARs whose development involves the WUSCHEL-RELATED HOMEOBOX11 (WOX11) regulatory pathway, have been previously shown to be induced as a response to wounding and callus formation in the roots (Ikeuchi et al., 2019; Sheng et al., 2017). AR formation interferes with the generation of a symmetric split-root system of shared ontogeny needed for our experiments. Therefore, we decided to investigate the impact of cutting instruments on the frequency of AR formation. As using a scalpel frequently resulted in the remaining main root stumps being pressed into the agar, for a subset of plants cut with a scalpel we decided to use tweezers to put the root stump back on top of the agar.

We found that the cutting instrument and approach significantly influenced the percentage of AR formation at the wound site 3 days after cutting (Figure 6). While using a razor blade may appear to be a 'cleaner' method of making a cut due to its sharpness and

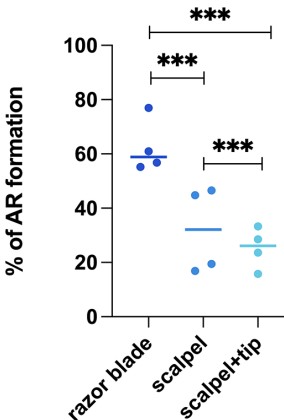

**Figure 6.** The percentage of plants with at least one adventitious root appearing at the location of the cut 3 days after cutting. *Arabidopsis thaliana* seedlings were grown for 7 days in d-root system before the cut. Each dot represents the mean of an independent replicate, where each independent replicate consists of $n$ = 10 plates with 6-7 plants per plate. First the mean was computed per plate, after which the mean of these means was computed over the 10 plates making up a single replicate. Lines show the mean of the group. The significance was calculated using chi-square test. *** $P<0.001$.

precision, it promoted a higher AR percentage. We hypothesize that the different tools used for cutting result in varying degrees of damage to the roots, with less damage leading to more frequent AR formation. Indeed, the lowest frequency of AR formation occurred after cutting with a scalpel and manipulating the root with tweezers. Since we aimed to avoid AR formation, we decided to cut our roots with a scalpel and lift the remaining main root stump with a tweezer for our final protocol as well as in the experiments shown in Figures 3–5 and discussed in the previous sections.

## 3. Discussion

In this article, we have explored the differences between replicability, reproducibility, and robustness. As experimental scientists, we aim to observe quantitatively and statistically similar effects when we replicate the same experiment (i.e. replicability) (Barba, 2018), with the possibility of full reproducibility (quantitatively identical results) reserved for computational analyses. However, when reporting experimental results, we place less emphasis on the robustness of our outcomes. Yet, reporting on the robustness or sensitivity of our outcomes to variation in the different relevant parameters is a well-established and expected practice in other domains, such as computational science (Confalonieri et al., 2010; Grimm & Berger, 2016). Communicating also in the experimental sciences which parameters of our protocols required optimization, and which were simply not varied would be very useful to fellow scientists interested in adopting the protocol. In addition, factors like seed quality (including mother plant growth conditions), germination rates and conditions, and further details such as light quality (not just intensity), etc. will all have a potentially massive impact on the protocols but generally are not detailed in our articles. It would not just save time, but it would also provide an understanding of which parameters interact with the outcomes of the experiments or the functioning of a protocol. We acknowledge that in collaborations with commercial parties, the strive for maximal transparency may collide with the request for confidentiality privileges by companies. We urge journals to take

a leading role in this matter, by expanding their requirements for reporting experimental methods.

Here we used Arabidopsis split-root nitrate foraging assays on plates as a case study on replicability and robustness of complex multi-step plant science protocols. Different variants of this assay have been documented in numerous articles over the past 15 years, with relatively similar outcomes of roots showing preferential foraging in the high nitrate compartment under heterogeneous conditions. Nonetheless, there is considerable diversity among split-root protocols associated with various protocol parameters, such as number of transfer steps, plant age, light intensity, and sucrose concentration in the medium (Table 1). It is currently unclear whether all these protocols are able to reproduce the complete range of root system responses presented in Ruffel et al., 2011. This highlights the importance of understanding and reporting on the possible extent to which outcomes are robust or sensitive to changes in protocol parameters.

The current study identified two parameters to which experimental outcomes are robust. The first is the root system compartment used in measurements, total, newly grown, or previously formed. We demonstrated that for the total and average of theLRs length, the outcome is independent of the section we examined (Figure 4). As expected, for the LRs number, which logically can mostly be affected in the newly growing root section after transfer to the split medium, outcomes when using the new section or overall are different from those using the old section. We recommend taking these findings into account, adjusting the root section measure to the type of root phenotype (LR length or rather number) one aims to investigate. The second parameter to which our experimental findings were found to be robust is the age of the plant during the treatment. In the existing literature, plant age at the end of the treatment ranges from 14 DAG (Remans et al., 2006), 15-20 DAG (Girin et al., 2010; Mounier et al., 2014; Ohkubo et al., 2017; Tabata et al., 2014), to a maximum of 23 DAG (Poitout et al., 2018; Ruffel et al., 2011). Exposing 15 and 20 DAG seedlings to the same treatment, we demonstrated that plant nitrogen foraging response is robust against changes in seedling age in our protocol (Figure 3).

In contrast to these parameters against which our experimental outcomes proved robust, we also identified parameters that did significantly affect outcomes. First, when roots were exposed to light instead of being covered by a D-root system, the foraging phenotype (where low-nitrogen-grown LRs are more and/or longer than high-nitrogen-grown LRs) observed in the new section of the root grown under homogenous low-nitrogen was lost (Figure 5). Even though in most split-root assays the root systems of plants are exposed to light, we developed the protocol using a D-root system, and it is possible that some of the optimizations are sub-optimal under conditions where roots are exposed to light. We currently do not know which of the steps of the protocol would be light-sensitive, or if light effects are dependent on the light source, light composition, photoperiod, and/or intensity. What is clear from our results is that the optimization done using the D-root system resulted in the loss of the foraging phenotype when the roots were exposed to light. As a second example of a parameter change leading to non-robust outcomes, we demonstrated that the cutting instrument, which is rarely mentioned in protocols, can have a significant effect (Figure 6). This highlights the importance of not only mentioning details of the major equipment like microscope type but also of the simple instruments used when writing protocols.

Based on the above discussion, we propose implementing more complete reporting conventions when describing experimental protocols, especially regarding experimental parameters that have been varied, and either required optimization or rather were found to have limited effect on obtaining the reported results. An example of a potential tool for writing experimental methodology would be a GitHub-type portal format. Just like the actual GitHub portal, which serves for storing codes and their versioning, in this portal, protocols can be detailed with thorough descriptions provided of any steps or parameters that were adjusted. Being interactive, this portal enables others to review, comment, and modify protocol steps, while maintaining a trackable record. In a separate section, peers can upload modifications they applied and notes on whether those modifications produced robust outcomes or not. This would pave the way to more collaborative and transparent scientific reporting within the research community and would also enhance the efficiency of studies that follow up on published work, which is the essence of the scientific process.

**Open peer review.** To view the open peer review materials for this article, please visit http://doi.org/10.1017/qpb.2025.10004.

**Supplementary material.** The supplementary material for this article can be found at http://doi.org/10.1017/qpb.2025.10004.

**Data availability statement.** The data that supports the findings of this study are available in the supplementary material of this article https://doi.org/10.5281/zenodo.14010234.

## Acknowledgements

We would like to thank Pierre Gautrat for guidance in experimental setup and fruitful discussions, Sara Buti and Muthanna Biddanda Devaiah for the technical advice and support, Benjamin Planterose Jimenez for his advice on statistics, and Otto van de Beek and his workshop at Utrecht University for the construction of the guillotine device. We acknowledge the use of Biorender, and Adobe Photoshop (version 25-12-0) for the creation of the graphic abstract and the use of Grammarly (version 1.87.1) and ChatGPT-4 for checking grammar and sentence structures.

**Author contributions.** K. ten Tusscher, L. Salvatore and R. Pierik conceived and designed the study. L. Salvatore conducted data gathering and performed statistical analyses. L. Salvatore, K. Kajala, K and ten Tusscher wrote the article. R. Pierik provided feedback on the article.

**Funding statement.** LS and KtT are supported by a VICI grant (VI.C.202.011 12268) and KK by a VIDI grant (VI.Vidi.193.104), both awarded by the Netherlands Organization for Scientific Research (NWO).

**Competing interest.** The authors declare none.

**Statistical analysis.** To avoid dependent data in the homogenous treatments (HNHN and LNLN) we took an average of the measurements of the paired roots. To determine the distribution of our data we used either a Shapiro-Wilk test, for datasets smaller than $n = 50$, or a D'Agostino-Pearson test for bigger datasets. As we couldn't prove normal distribution we opted to use non-parametric test to evaluate the significance of our comparisons. Most of the comparisons were evaluated with a Mann-Whitney test, except for the comparison between HNln vs LNhn, where due to the data being paired, we used a Wilcoxon test.

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
