## [Reviewer Report]

This was an interesting and enjoyable paper to read.

I couldn’t agree more with the authors' points on scientific reporting.

I hope this manuscript encourages experimental scientists to be more thorough within their reporting. The idea that experimentalists should report on robustness is beautiful, and obviously a good idea once it has been said.

The experimental system chosen to illustrate the authors points was well chosen, simple enough for scientists from all fields to understand, complex enough to be a useful illustration tool.

There are three minor points regarding the manuscript.

1) The significance information in the boxplots in figures 3, 4 and 5 would be clearer in table form. I think the figures do not need to be removed or edited, they are informative. I think the addition of tables to the manuscript showing significance measures is important to enable the reader to follow the discussions, and to show the significance data. The significance tables may be large, if they do not fit in the main manuscript, please put them in the supplementary information and point to them in the manuscript.

2) The sentence on lines 130 to 133 was hard to understand. It is such an important point for the paper that I think the text needs to be rewritten, more clearly.

3) Figure 2 was very attractive, but it took me a while to work out what it was showing. It is a lovely figure which does show what the authors' intent. I think the figure could be explained a little more clearly, all that is needed is for the figure legend to explain that the black typed text is the original protocol (as is described on line 149). Would be nice to tell the reader in the legend that the protocol modifications in the figure are discussed in the following sections.

---

## [Reviewer Report]

In “Ensuring Robustness in Scientific Research, Split-root assays as an example case”, the authors show how a more detailed description of protocols could increase replicability of research and robustness of conclusions. In particular, they plead for including a description of which parts of the protocols have been optimized and which not; which parts of the protocol are critical; which “simple” materials have been used; how much time and how many initial seeds (etc) are needed.

They use the split root assay as an example of a complex protocol with substantial variation within the field, and demonstrate that some factors matter a lot, whereas others can be varied without affecting conclusions.

I think this is an important topic for quantitative (plant) biology. The article is well written. I have overall few comments, except that for a paper that aims at setting a methodological example, the statistical methods are not so well described and may need improvement (see below). The complementary star methods give a good example of a protocol as advocated.

I would like to challenge the authors to think big on what they propose, and see if they can address the following. The split root assays probably are uniquely used in fundamental science, so supporting maximum replicability should always be in the author’s interest. In some cases, however, there may be a conflict of interests in some wanting to exploit advantage of having an optimized protocol in house (e.g., for a recalcitrant species, in a project with a plant breeder on board, etc). Can this work be generalized to think of both (increased) minimum and optimal reporting standards/methods for protocols, or is this only a call to goodwill?

Lines 271-271: “These results demonstrate that seedling age, at least within the range tested here, does not significantly affect the nitrogen-foraging phenotypes of interest.”

Judging from the markers for statistical significance in figure 3, there seems to be a difference. Although visually, the trends look the same for both 15 DAG and 20 DAG, not all comparisons are significantly different at the alpha = 0.05 level for 20 DAG or 15 DAG. This difference is not even acknowledged in the main text, which I think is a bad thing. The authors could note this and state that there might be a difference in power of the assay depending on the length of the protocol, but establishing such a difference with certainty would require additional repetitions beyond the scope of this manuscript.

Statistical testing:

I am highly surprised that paired ratio t-tests and one-way ANOVA are used for the count data (LR number), whereas the non-parameteric Wilcoxon and Kruskal-Wallis tests are used for the continuous (LR length) data. I could see the rationale of doing it the other way around, i.e., considering that count data are by definition not normally distributed (Poisson perhaps, or a different discrete distribution). Was this choice of methods the result of some testing for normality of the distributions (without reporting)? Whatever the motivation was, it would be good to have it reported.

Multiple legends state “the rest of the comparisons were analyzed using one-way ANOVA”. As this ANOVA test only tells whether there is a difference among the groups, but not the pairwise differences, most likely a post hoc test has been used. If yes: which one? If no: what exactly has been tested? If a different one-way ANOVA has been used to compare each pair of (HNHN vs HNln, LNhn vs LNLN, HNHN vs LNLN), than this would be the equivalent of three t-tests. In total, 4 t-tests are then performed on the sample, so some correction for multiple comparisons would need to be applied. It is not reported whether or not this is done.

Similar applies to where Kruskal-Wallis tests are performed.

It is unclear to me how many roots are in the HNHN and LNLN samples. Does n=52-55 refer to the number of plants, or the number of roots? If plants: are the quantities first averaged per plant (left/right), or are the ~100 roots added individually to the plot, or only the left (or right) roots on the plate?

Degrees of freedom are not reported.

Minor comments:

Figure 1: it looks like the plot shows real data (but without a y-axis or -label). Although not important for the story, it would be good to state which data is shown.

Multiple figures: Please label “total LR length” where applicable, to make this immediately obvious (vs average).

Line 268: average LR length is in figure 3D+F.

Figure legends in general: I think it would be a good signal to the readers to add to each legend some information that varied within the manuscript, such as whether data was measured on the new part / total root; roots were grown under light or dark conditions.

Figure S5: image missing/broken.

---

## [Editor Report]

I apologise for the significant time the reviews took. The comments of the reviewers are positive and they acknowledge the significance of the topic. Following the suggestions of the reviewers I recommend Minor Revision. I would urge the authors to be more precise regarding the statistical methods and provide a better justification for the statistical methods applied.

---

## [Reviewer Report]

The authors have addressed my comments. Thank you.

However, I do not like the image which has been added to page 1.

Minor revision suggested - but not essential, as it could be a matter of taste.

The AI generated image of a young scientist frowning/sulking while holding two split root plates is cartoon like and clownish. In my opinion the image undermines the scientific importance of the work in the manuscript.

The frown/sulk also gives a negative impression of the work to follow.

Apart from the split root plates the image does not inform the reader about the content of the paper.

I advise that, if an image is needed, it is an informative and positive (emotionally neutral) image is generated.

I’d remove any people from the image, for inclusivity reasons.

---

## [Editor Report]

The authors are happy with the revision. However, one reviewer comments on the graphical abstract. I agree with this comment and advise the authors to either remove it or replace it by something more neutral and informative.

---

## [Editor Report]

Dear Kirsten,

I find the graphical abstract now more suitable for a scientific publication and I am happy to finally formally accept the manuscript for publication. 

Christian